# Intratumoral Cytotoxic T-Lymphocyte Density and PD-L1 Expression Are Prognostic Biomarkers for Patients with Colorectal Cancer

**DOI:** 10.3390/medicina55110723

**Published:** 2019-10-31

**Authors:** Ilknur Calik, Muhammet Calik, Gulistan Turken, Ibrahim Hanifi Ozercan, Adile Ferda Dagli, Gokhan Artas, Burcu Sarikaya

**Affiliations:** Department of Pathology, Faculty of Medicine, Fırat University, 23000 Elazığ, Turkey; drmuco2011@gmail.com (M.C.); gulistanturken1@gmail.com (G.T.); ozercanih@gmail.com (I.H.O.); ferda58@yahoo.com (A.F.D.); gartas79@gmail.com (G.A.); burcuburcuyzc@gmail.com (B.S.)

**Keywords:** cytotoxic T lymphocytes, colorectal cancer, PD-L1 expression, prognosis, tumor microenvironment

## Abstract

*Background and objectives:* Cytotoxic T-lymphocyte (CTL)-mediated inflammatory response to tumors plays a crucial role in preventing the progression of some cancers. Programmed cell death ligand 1 (PD-L1), a cell-surface glycoprotein, has been reported to repress T-cell-mediated immune responses against tumors. However, the clinical significance of PD-L1 in colorectal cancer (CRC) remains unclear. Our aim was to elucidate the prognostic significance of PD-L1 expression and CD8+ CTL density in CRC. *Materials and methods*: CD8 and PD-L1 immunostaining was conducted on 157 pathologic specimens from patients with CRC. The CD8+ CTL density and PD-L1 expression within the tumor microenvironment were assessed by immunohistochemistry. *Results:* Tumor invasion (pT) was significantly correlated with intratumoral (*p* = 0.011) and peritumoral (*p* = 0.016) CD8+ CTLs density in the tumor microenvironment. In addition, there was a significant difference in the intensity of CD8+ CTLs between patients with and without distant metastases (intratumoral *p* = 0.007; peritumoral *p* = 0.037, T-test). Lymph node metastasis (pN) and TNM stage were significantly correlated with PD-L1 expression in CRC cells (*p* = 0.015, *p* = 0.029, respectively). Multivariate analysis revealed a statistically significant relationship between the intratumoral CD8+ CTL density and disease-free survival (DFS) (hazard ratio [HR] 2.06; 95% confidence interval [CI]: 1.01–4.23; *p* = 0.043). The DFS was considerably shorter in patients with a high expression of PD-L1 in cancer cells than those with a low expression (univariate HR 2.55; 95% CI 1.50–4.34; *p* = 0.001; multivariate HR 0.48; 95% CI 0.28–0.82; *p* = 0.007). Conversely, patients with high PD-L1 expression in tumor-infiltrating lymphocytes had a longer DFS in both univariate analysis (HR 0.25; 95% CI: 0.14–0.44; *p* < 0.001) and multivariate analysis (HR 3.42; 95% CI: 1.95–6.01; *p* < 0.001). *Conclusion*: The CD8+ CTL density and PD-L1 expression are prognostic biomarkers for the survival of patients with CRC.

## 1. Introduction

Colorectal cancer (CRC), one of the most common malignancies, is a major contributor to cancer-related deaths worldwide, and its incidence is increasing in developing countries. It is estimated that in 2018 there will be more than 1.8 million new colorectal cancer cases and 881,000 deaths [1,2,3,4]. Only prevention by endoscopy or by noninvasive screening has been shown useful to reduce incidence and prevalence of CRC [5,6]. The surgical procedures and chemotherapy for CRC are improving, but patient prognosis is still poor [7]. Physicians generally use a tumor-node-metastasis (TNM) staging system to estimate the prognosis of this cancer, but patients with the same stage or histologic grade frequently show nonhomogeneous biological behaviors. Furthermore, despite the use of classical prognostic parameters, such as tumor site, tumor size, histopathologic type, grading, and TNM staging, no credible prognostic systems yet exist for CRC. Consequently, defining other biomarkers would be helpful for developing dependable prognostic procedures for CRC [7,8].

Recent researches have indicated an interrelationship between the host-inflammatory response and carcinogenesis. The inflammatory reaction that arises against tumors has a crucial role in determining the occurrence, progression, and dissemination of some cancers [7,8,9]. The presence of tumor-infiltrating lymphocytes (TILs), and especially CD8+ cytotoxic T lymphocytes (CTLs), is correlated with the immune status of the body, and a variety of studies have identified the CTLs intensity as a favorable biomarker for the prognosis of many cancers, including CRC [10,11,12]. In many tumors, CTLs serve as gatekeepers to prevent tumor spread. However, tumors seldom disappear spontaneously, because of their capability to form an immunosuppressive microenvironment by activating immune control points, such as programmed cell death 1 ligand 1 (PD-L1) [13,14].

PD-L1 is a cell-surface glycoprotein member of the B7 family that adversely regulates T lymphocytes to cause exhaustion of lymphocyte numbers via the programmed cell death 1 receptor (PD-1) [15]. PD-L1 is expressed by various inflammatory cells and is upregulated through many inflammatory mediators and cytokines, such as interferon-gamma (IFN-γ). Upregulation of PD-L1 in cancer cells suppresses CTLs’ effectiveness. PD-L1 is also expressed in many types of tumors, such as lung, bladder, ovarian, esophageal, and renal cancer, and its expression is associated with a debilitated host immune reaction and unfavorable patient prognosis [15,16,17]. However, the effectiveness of PD-L1 expression as a prognostic indicator has not yet been adequately demonstrated in patients with CRC.

The aim of this study was to investigate the effect of PD-L1 expression and the CTLs’ response of the host on the prognosis of patients with CRC. We also evaluated the relationship between these factors (the PD-L1 expression and CTLs density) and classical clinicopathological parameters, such as tumor site, histopathologic type, grading, depth of tumor invasion (pT), lymph node metastasis (pN), and TNM stage.

## 2. Materials and Methods

### 2.1. Patients and Pathological Specimens

This study was approved by the Firat University Ethical Committee (Date: 17 September 2019, Approval No: 13-08). We retrospectively evaluated pathological specimens of 157 patients who had undergone surgery for CRC between 2011 and 2014, at Firat University Hospital. Patients who were treated with chemotherapy were not included in the study. We enrolled 18 patients with TNM stage I, 40 with stage II, 80 with stage III, and 19 with stage IV. A control group (n = 157) consisting of non-tumoral colorectal tissues from the same patients was included in the study. Two pathologists (Calik I and Calik M) histologically re-evaluated each pathologic material. The clinical and pathological data were acquired from hospital medical and pathologic reports. The TNM stages of the cases were specified according to the American Joint Committee on Cancer (AJCC), 7th edition. Survival data included patient outcome and the interval between the date of surgical resection and the date of death.

### 2.2. Immunohistochemistry

Immunohistochemistry (IHC) was performed using histological tissue microarray slides that were 3 µm thick. The following antibodies were used: anti-PD-L1 (clone CAL10, Master Diagnostica, Granada, Spain) and anti-CD8 (SP57, Ventana, AZ, USA). The sections were stained using the Ventana BenchMark Ultra Autostainer (Ventana, Tucson, AZ-85755, USA) and the UltraView Universal DAB kit (Ventana, Tucson, AZ-85755, USA), following the manufacturer’s instructions. PD-L1 expression in cancer cells (CCs) and TILs, as well as the density of CTLs, were evaluated by IHC. When evaluating PD-L1 expression, non-tumoral colorectal mucosa were utilized as internal negative controls. Tonsillectomy materials in the pathology archive were used as positive controls.

### 2.3. Scoring System for PD-L1 Expression

PD-L1 expression in CCs and TILs was semi-quantitatively scored according to the intensity and diffuseness of staining, using the following scale (0–3+): 0, absent; 1+, weak; 2+, moderate; 3+, strong membrane staining. PD-L1 expression intensity was also scored as low when <5% of the cells were PD-L1 positive and high when ≥5% of the cells were positive. These criteria had previously been validated in various types of cancers [2,18].

### 2.4. Scoring System for CD8+ CTLs Density

The intratumoral CD8+ (I-CD8+) CTLs density was scored as low when the mean CTLs number was <50 and high when it was ≥50. The density of the peritumoral CD8+ (P-CD8+) CTLs was scored as low when their mean number was <200 and high when it was ≥200. CTLs numbers were counted twice in a microscope field at ×200 magnification. These criteria had been previously validated in studies on CRC [2,11,19].

### 2.5. Statistical Analyses

The data were analyzed statistically, using SPSS v.20 software and were expressed as percentages, means, and standard deviations. The normal distribution of the data was evaluated with the Shapiro–Wilk test. The *p*-values > 0.05 were accepted as indicating a normal distribution. Kurtosis and skewness values between −2 and +2 were also considered to indicate a normal distribution. The Pearson test was used to investigate the relationship between normally distributed data. An independent sample t-test and ANOVA were used to identify variances between the groups. The chi-square test was used to determine the relationship between data that were not normally distributed. The relationships between overall survival (OS)/disease-free survival (DFS) and PD-L1 expression and CTLs density were evaluated using the Kaplan–Meier method (log-rank test). Cox regression analysis was applied to estimate the hazard ratios (HRs) and 95% confidence intervals (Cis) for univariate and multivariate models. The *p* < 0.05 threshold was considered statistically significant for all data.

## 3. Results

### 3.1. General Clinicopathological Features of the Cases and Their Relationship with Disease-Free Survival

A total of 157 patients were included in the present study. The case characteristics are summarized in Table 1. The median follow-up of all patients was 52.7 ± 13.6 months. Overall, 100 (63.7%) patients experienced recurrence. No differences were noted in sex, tumor size, or histologic grade in terms of DFS, whereas statistically significant correlations were noted for age, tumor site, histopathologic type, pT, pN, distant metastasis, and TNM stage versus DFS (Table 1).

### 3.2. Relationship of CTLs Density to Classical Clinicopathologic Features

Examination of the cases revealed 93 (59.2%) patients with low and 64 (40.8%) with high I CD8+ CTLs densities (Figure 1A,B). The P-CD8+ CTLs densities were low in 95 (60.5%) patients and high in 62 (39.5%) patients (Figure 1C,D). Table 2 shows that high I-CD8+/P-CD8+ CTLs density in tumors was associated with both pT and distant metastasis. However, no relationship was evident between CTL density and either pN or TNM stage.

### 3.3. Relationship of CTLs Density to OS and DFS

The univariate analysis showed a considerably longer DFS in patients with high I-CD8+ CTLs and P-CD8+ CTLs densities than with low densities (Table 3). The multivariate analysis did not reveal any correlation between P-CD8+ CTLs density and DFS (HR 0.76; 95% CI: 0.38–1.53; *p* = 0.459). However, a statistically significant relationship was evident between I-CD8+ CTL density and DFS (HR 2.06; 95% CI: 1.01–4.23; *p* = 0.043). Evaluation of the relationship of I-CD8+ CTL and P-CD8+ CTL density with OS revealed longer OS in patients with higher ICD8+/P-CD8+ CTL density (Log-rank, *p* = 0.002; *p* = 0.011, respectively) (Figure 2A,B).

### 3.4. Relationship of PD-L1 Expression to Classical Clinicopathological Features

PD-L1 was not expressed in normal colon mucosa (Figure 3A,B). An examination of the PD-L1 expression in CCs revealed low expression in 85 (54.1%) patients and high expression in 72 (45.9%) patients (Figure 3C,D). The expression in TILs was low in 72 (45.9%) patients and high in 85 (54.1%) patients (Figure 3E,F). The high PD-L1 expression in CCs was closely correlated with the histologic type (*p* < 0.001), pT (*p* < 0.001), pN (*p* = 0.015), and TNM stage (*p* = 0.029). Similarly, a relationship was evident between PDL1 expression in TILs and histologic type (*p* = 0.038), pT (*p* = 0.026), pN (*p* < 0.001), and TNM stage (*p* = 0.032). A significant correlation was also detected between PD-L1 expression and CD8+ CTLs density. However, no association was noted between PDL1 expression on TILs and patient sex, age, tumor site, or histologic grade (Table 4).

### 3.5. Relationship of PD-L1 Expression to OS and DFS

The Kaplan–Meier survival analysis showed a significantly shorter DFS/OS in patients with high PD-L1 expression in CCs than with low expression (Log Rank, *p* < 0.001), while DFS/OS was higher in patients with high PD-L1 expression in TILs than with low expression (Figure 4A,B). The mean overall survival was 45.21 ± 1.94 months in patients with low PD-L1 expression in TILs and 59.15 ± 0.49 months in patients with high expression. The DFS was also significantly shorter in patients with a high PD-L1 expression in CCs than with low expression, according to both univariate (HR 2.55; 95% CI: 1.50–4.34; *p* = 0.001) and multivariate (HR 0.48; 95% CI: 0.28–0.82, *p* = 0.007) analyses. Conversely, DFS was substantially longer in patients with high PD-L1 expression in TILs than with low expression, according to both univariate (HR 0.25; 95% CI: 0.14–0.44; *p* < 0.001) and multivariate (HR 3.42; 95% CI: 1.95–6.01; *p* < 0.001) analyses (Table 3).

## 4. Discussion

It is extremely important to establish exact predictive systems or biomarkers in identifying low- and high-risk groups and improving suitable treatment modalities for patients with CRC. Recent areas of focus have included different immunotherapies, expression of PD-L1, and the density of TILs in various human malignancies [20,21,22,23,24]. A correlation between PD-L1-expression and TILs, and especially CTLs, is not yet fully established for CRC. The aim of this study was to identify prognostic roles for CTLs density and PD-L1 expression by immunohistochemistry, using human CRC tissues.

Since the 1800s, when Rudolf Virchow first described the presence of inflammatory cells in tumor sites and put forward the theory that tumors occur in areas of chronic inflammation, it has attracted the attention of scientists trying to understand the basics of inflammation and the biology of cancer. It is known that the risk of developing colorectal cancer is higher in patients with inflammatory bowel disease (Ulcerative colitis and Crohn’s disease) than in the normal population. Despite the knowledge that chronic inflammatory processes can be a triggering factor for cancer development, many experimental and retrospective studies emphasize the protective role of immune cells in cancer development [25]. Tumor microenvironments constitute an important step in the presentation of tumor antigens to T cells, and many immuno-inflammatory cells contribute to the tumor development process [11]. In addition, a recent study demonstrated that T cells are part of the immune response to the tumor and that immuno-inflammatory cells, particularly T lymphocytes, infiltrating the tumor tissue can predict CRC-specific survival [26]. In another study, CD8+ T lymphocyte density was strongly associated with positive clinical outcomes in patients with CRC and predicted survival time more effectively than histopathology-based staging [27]. Intratumoral infiltration of T cells was found to be an important predictor of survival outcome in rectal cancer. Immuno-inflammatory cells have an important role in the suppression of host antitumor immunity and tumor-cell migration and invasion. Previous studies by Klintrup et al. [28] and Huh et al. [29] reported a correlation between a greater number of inflammatory cells and favorable prognosis in CRC. Trajkovski et al. [12] showed a significant correlation between a high number of CD8+ CTLs and good prognosis. Despite these evident correlations between high CD8+ CTLs density and survival in CRC, the effect on other clinicopathological parameters has not been fully investigated [30,31]. In the present study, a high CD8+ CTLs density was significantly correlated with better survival. Furthermore, patients with high CTLs density had significantly more favorable tumor behaviors, such as lower pT and distant metastasis, when compared with patients with low density. In agreement with the findings of Lee et al. [31], the DFS rate in the present study was better in patients with high CD8+ CTLs density than with low density. Our univariate analysis confirmed high DFS rates in patients with high I-CD8+ CTLs density and high P-CD8+ CTLs density; however, our multivariate analysis indicated that only patients with high I-CD8+ CTLs density had a longer survival rate (Table 3). These findings in our series indicate that the density of CD8+ CTLs can provide important prognostic knowledge about patients with CRC.

The interaction between PD-L1 and PD-1 causes an exhausted phenotype and dysfunction of T cells [32]. The PD1/PD-L1 signaling pathway is a negative feedback system that inhibits the activity of T cells [2,16,32]. PD-L1 overexpression has been demonstrated in cancer cells or tumor-infiltrating immune-inflammatory cells in many malignant neoplasms, including CRC. PD-L1 may play a central role in immuno-oncological interactions [33]. Various different mechanisms have been asserted for PD-L1 upregulation in tumor cells: (1) innate intrinsic induction—constitutive oncogenic signaling in tumor cells, such as ALK and EGFR, leading to overexpression of PD-L1; and (2) adaptive immune resistance—stimulation of PD-L1 expression in tumor cells in reply to local inflammatory signals produced by the active immune response, such as CD8 cytotoxic T lymphocytes [34]. Recent studies have discovered the relationship between PD-L1 expression and immune cell infiltration in the tumor microenvironment. IFN-γ secreted by the infiltrated CD8+ CTLs was required for PD-L1 induction, indicating that overexpression of PD-L1 within the tumor microenvironment functioned as a negative feedback mechanism, which represents a compensatory immune response by CD8+ CTLs and IFN-γ within the tumor microenvironment [34,35].

Previous studies have shown controversial conclusions about whether PD-L1 expression indicates a better or worse prognosis in CRC, probably because of differences in study populations and designs [33,36]. For example, Droeser et al. demonstrated an association between PD-L1 expression and better prognosis in CRC [37], whereas another study reported a worse CRC prognosis in association with PD-L1 expression in CCs and TILs [38]. Conversely, Koganemaru et al. [2] found a significant association between high PD-L1 expression in CCs and poor prognosis, whereas high PD-L1 expression in TILs was associated with a good prognosis. Some researchers asserted that the prognostic implication of PD-L1 expression might depend on the mismatch repair (MMR) status, but the outcomes were incoherent [37,38]. CRCs with MMR defects have high microsatellite instability (MSI) and account for about 12% to 15% of all cases [33]. These tumors contain high-density TILs and are characterized by a better prognosis compared to those without MMR defects. Some studies have stated different effects of PD-L1 expression on prognosis according to MSI status. Dunne et al. [39] proved that PD-L1 expression was associated with a significantly worse DFS in MMR-deficient tumors. However, PD-L1 expression did not demonstrate a statistically significant correlation with DFS in patients who were not MMR-deficient. Conversely, Droeser et al. [37] demonstrated that high PD-L1 expression in MMR-proficient CRC was correlated with early tumor stage, absence of lymph node metastases, lower histological grade, absence of vascular invasion, high numbers of TILs, especially CD8+ T cells, and an improved overall survival rate. Unfortunately, we did not assess the status of germline mutations in MLH1, MSH2, and MSH6. Therefore, further studies are needed to evaluate the possible interrelation between PD-L1 expression and MSI status.

In the present study, PD-L1 expression on TILs was independently correlated with improved oncologic outcomes, which was in direct contrast to PD-L1 expression in CCs. However, contrary to our data, Wang et al. reported that PD-L1 positivity on TILs was an indicator of poor prognosis in CRC [40], although another recent study indicated that PD-L1 expression on TILs positively affected the survival of patients with CRC [33]. When we evaluated the recurrence patterns according to PD-L1 expression, we found a significantly higher systemic metastasis rate in the low PD-L1 expression group than in the high PD-L1 expression group. In addition, low PD-L1 expression was highly correlated with hepatic recurrence. The high systemic recurrence rate and a significant correlation with hepatic recurrence in the low PD-L1 expression group may reflect worse biological behavior. In the present study, both OS and DFS rates were significantly lower in patients with high PD-L1 expression in CCs than with low expression. In addition, cases with high PD-L1 expression in CCs showed more lymph node metastasis, and their TNM stage was more advanced.

The immune microenvironment status, as indicated by PD-L1 expression in CCs and CTLs, may provide potential predictive biomarkers in cancer immunotherapy [2,16]. Lately, novel therapies using immunological control point inhibitors (e.g., PD-L1 inhibitors) have achieved groundbreaking results in various types of cancers [2,15,16,32]. Immunological control points are one of the steps in immune avoidance and tumor development; therefore, the use of monoclonal antibodies to inhibit the molecules associated with these control points restores the host immune response, thereby preventing tumor growth and even facilitating tumor regression. Therefore, immunotherapy using immunological checkpoint inhibitors is a rapidly growing modality for the treatment of certain human cancers [14,15,16]. As seen in our study, increased PD-L1 expression in CCs suppresses CTLs’ density and negatively affects prognosis.

On the other hand, our study had some limitations. First, it was a retrospective, single-institution study, so the potential exists for selection bias. Second, the cut-off value for PD-L1 expression varied between studies; therefore, future researchers should strive for standardization of the methods used to detect PDL1 expression. Third, we focused only on CD8+ CTLs density; however, other studies have shown PD-L1 expression in various myeloid cells in CRC [2]. Further research is needed to identify the properties of PD-L1 expression in other inflammatory cells and to determine the function of PD-L1 in these cells.

## 5. Conclusions

In conclusion, our findings showed that high CD8+ CTL concentration correlated significantly with good tumor behavior and better patient survival. There was an inverse correlation between the density of CTLs and PD-L1 expression; especially in patients with high PD-L1 expression in CCs, CTLs density was low. These patients had low OS and DFS rates. PD-L1 expression was found to be significantly associated with multiple lymph node metastases and advanced TNM stage. In view of these results, CTLs density and PD-L1 expression can be used as prognostic markers in patients with CRC. Furthermore, we expect that future studies on PD-L1 expression and tumor inflammatory responses may contribute to the development of new CRC therapies.

## Figures and Tables

**Figure 1 medicina-55-00723-f001:**
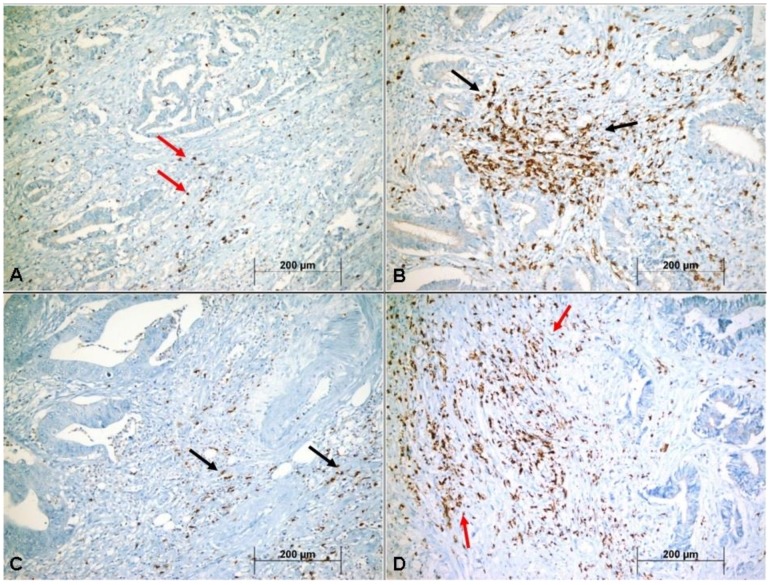
Representative images showing low density of intratumoral CD8+ (I-CD8+) cytotoxic T lymphocytes (CTLs) (red arrows) (**A**), high density of I-CD8+ CTLs (black arrows) (**B**), low density of peritumoral CD-8+ (P-CD8+) CTLs (black arrows) (**C**), and high density of P-CD8+ CTLs (red arrows) (**D**) in colorectal cancer (×400).

**Figure 2 medicina-55-00723-f002:**
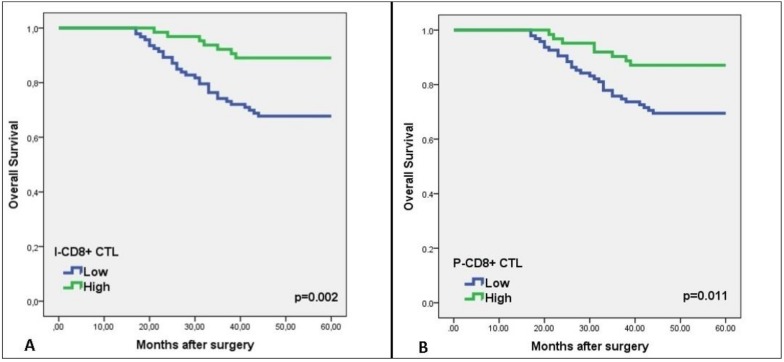
Kaplan–Meier curves of overall survival (OS) versus CD8+ cytotoxic T lymphocytes (CTLs) density in patients with colorectal cancer. Kaplan–Meier curves demonstrating an association between high intratumoral CD8+ (I-CD8+) (**A**) and peritumoral CD8+ (P-CD8+) CTLs density (**B**) versus five-year OS (*p* = 0.002, *p* = 0.011, respectively).

**Figure 3 medicina-55-00723-f003:**
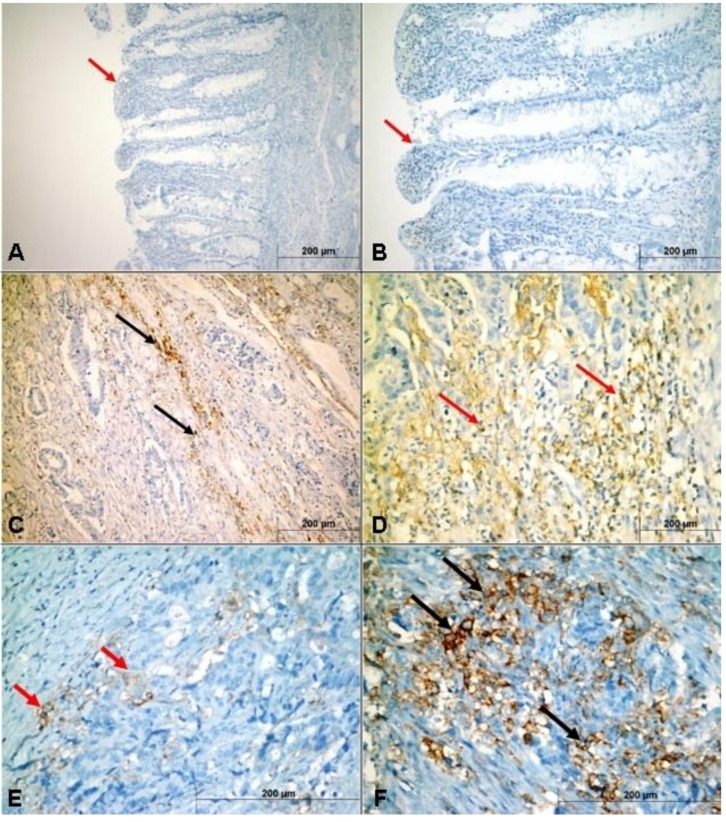
Expression patterns of PD-L1 within the tumor microenvironment of colorectal carcinoma. PD-L1 expression in normal colorectal mucosa (red arrows) (**A**,**B**). Low (black arrows) (**C**) and high (red arrows) (**D**) PD-L1 expression in cancer cells (CCs). Low (red arrows) (**E**) and high (black arrows) (**F**) PD-L1 expression on tumor-infiltrating lymphocytes (TILs).

**Figure 4 medicina-55-00723-f004:**
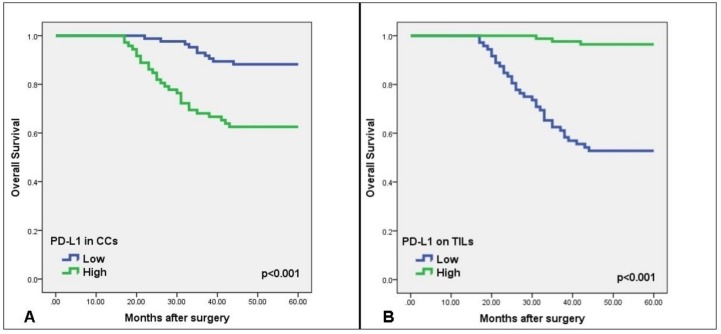
Kaplan–Meier curves of overall survival (OS) versus PD-L1 expression in patients with colorectal cancer. Kaplan–Meier curves demonstrating an association between PD-L1 expression by cancer cells (CCs) (**A**) and by tumor-infiltrating lymphocytes (TILs) (**B**) versus five-year OS (*p* < 0.001).

**Table 1 medicina-55-00723-t001:** General features of cases and correlation of clinicopathologic characteristics with disease-free survival (n = 157).

		Recurrence n (%)		Cox Regression (Univariate)		
	n (%)	Absent n (%)	Present n (%)	HR	95% CI	*p*-Value
Sex						
Male	97 (61.8)	59 (60.8)	38 (39.2)	0.82	0.48–1.42	0.497
Female	60 (38.2)	40 (66.7)	20 (33.3)			
Age (median 60)						
(range 19–90)						
19–44	25 (15.9)	21 (84.0)	4 (16.0)	3.14	1.12–8.84	0.029
45–54	50 (31.8)	33 (66.0)	17 (34.0)			
≥55	82 (52.2)	45 (54.9)	37 (45.1)			
Tumor site						
Right side	43 (27.4)	20 (46.5)	23 (53.5)	0.5	0.29–0.84	0.01
Left side	114 (72.6)	79 (69.3)	35 (30.7)			
Tumor size						
<5 cm	61 (38.9)	41 (67.2)	20 (32.8)	1.3	0.75–2.24	0.342
≥5 cm	96 (61.1)	58 (60.4)	38 (39.6)			
Histopathologic type						
Adenocarcinoma	99 (63.1)	69 (69.7)	30 (30.3)	0.38	0.18–0.80	0.012
Mucinous	38 (24.2)	21 (55.3)	17 (44.7)			
Signet-ring	20 (12.7)	9 (45.0)	11 (55.0)			
Histologic grade						
Well	25 (15.9)	16 (64.0)	9 (36.0)	0.58	0.29–1.13	0.114
Moderate	109 (69.4)	71 (65.1)	38 (34.9)			
Poor	23 (14.6)	12 (52.2)	11 (47.8)			
Depth of invasion						
pT1	26 (16.6)	20 (76.9)	6 (23.1)	0.42	0.24–0.75	0.004
pT2	64 (40.8)	46 (71.9)	18 (28.1)			
pT3	67 (42.7)	33 (49.3)	34 (50.7)			
Lymph node status						
Absent	91 (58.0)	67 (73.6)	24 (26.4)	0.51	0.25–1.02	0.008
1–3	34 (21.7)	12 (35.3)	22 (64.7)			
≥4	32 (20.4)	20 (62.5)	12 (37.5)			
Distant Metastasis						
Absent	123 (78.3)	90 (73.2)	33 (26.8)	0.15	0.09–0.27	<0.001
Present	34 (21.7)	9 (26.5)	25 (73.5)			
TNM staging						
Stage I	18 (11.5)	13 (72.2)	5 (27.8)	0.14	0.22–0.84	<0.001
Stage II	40 (25.5)	33 (82.5)	7 (12.5)			
Stage III	80 (50.9)	46 (57.5)	34 (42.5)			
Stage IV	19 (12.1)	7 (36.8)	12 (63.2)			

Hazard ratio, HR; 95% Confidence interval, 95% CI.

**Table 2 medicina-55-00723-t002:** Correlation of intratumoral CD8+ (I-CD8+) cytotoxic T cells (CTLs) and peritumoral CD8+ (P-CD8+) CTLs density with clinicopathologic features in patients with colorectal cancer (n = 157).

	I-CD8+ CTLs Density			P-CD8+ CTLs Density		
	Low n (%)	High n (%)	*p*-Value	Low n (%)	High n (%)	*p*-Value
Sex						
Male	60 (61.9)	37 (38.1)	0.442 †	62 (63.9)	35 (36.1)	0.270 †
Female	33 (55.0)	27 (45.5)		33 (55.0)	27 (45.5)	
Age (median 60)						
(range 19–90)						
19–44	15 (60.0)	10 (40.0)	0.992 ‡	14 (56.0)	11 (44.0)	0.856 ‡
45–54	29 (58.0)	21 (42.0)		30 (60.0)	20 (40.0)	
≥55	49 (59.8)	33 (40.2)		51 (62.2)	31 (37.8)	
Tumor site						
Right side	27 (62.8)	16 (37.2)	0.147 †	28 (65.1)	15 (35.9)	0.472 †
Left side	66 (57.9)	48 (42.1)		67 (58.8)	47 (41.2)	
Tumor size						
<5 cm	34 (56.7)	26 (43.3)	0.663 †	32 (53.3)	28 (46.7)	0.150 †
≥5 cm	59 (60.8)	38 (39.2)		63 (64.9)	34 (35.1)	
Histopathologic type						
Adenocarcinoma	55 (55.6)	44 (44.4)	0.055 ‡	55 (55.5)	44 (44.4)	0.117 ‡
Mucinous	21 (55.3)	17 (44.7)		24 (63.2)	14 (36.8)	
Signet-ring	17 (85.0)	3 (15.0)		16 (80.0)	4 (20.0)	
Histologic grade						
Well	14 (56.0)	11 (44.0)	0.884 ‡	14 (56.0)	11 (44.0)	0.600 ‡
Moderate	64 (58.7)	45 (41.3)		65 (59.6)	44 (40.4)	
Poor	15 (65.2)	8 (34.8)		16 (69.6)	7 (30.4)	
Depth of invasion						
pT1	9 (34.6)	17 (65.4)	0.011‡	12 (46.2)	14 (53.8)	0.016 ‡
pT2	38 (59.4)	26 (40.6)		34 (53.1)	30 (46.9)	
pT3	46 (68.7)	21 (31.3)		49 (73.1)	18 (26.9)	
Node status						
Absent	53 (58.2)	38 (41.8)	0.696 ‡	54 (59.3)	37 (40.7)	0.939 ‡
1–3	19 (55.9)	15 (44.1)		21 (61.8)	13 (38.2)	
≥4	21 (65.6)	11 (34.4)		20 (62.5)	12 (37.5)	
Distant Metastasis						
Absent	66 (53.7)	57 (46.3)	0.007 †	70 (56.9)	53 (43.1)	0.037 †
Present	27 (79.4)	7 (20.6)		25 (73.5)	9 (26.5)	
TNM staging						
Stage I	9 (50.0)	9 (50.0)	0.512 ‡	11 (61.1)	7 (38.9)	0.486 ‡
Stage II	23 (57.5)	17 (42.5)		21 (52.5)	19 (47.5)	
Stage III	47 (58.8)	33 (41.2)		49 (61.2)	31 (38.8)	
Stage IV	14 (73.7)	5 (26.3)		14 (73.7)	5 (26.3)	

† Independent simple T-test, ‡ one-way ANOVA.

**Table 3 medicina-55-00723-t003:** Cox regression analysis (univariate and multivariate) of cytotoxic T cells (CTLs) density and PD-L1 expression associated with disease-free survival.

Parameters	Univariate		Multivariate	
	HR (95% CI)	*p*-Value	HR (95% CI)	*p*-Value
I-CD8+ CTLs density	0.41 (0.23–0.75)	0.004	2.06 (1.01–4.23)	0.043
P-CD8+ CTLs density	0.39 (0.23–0.66)	0.001	0.76 (0.38–1.53)	0.459
PD-L1 in CCs	2.55 (1.50–4.34)	0.001	0.48 (0.28–0.82)	0.007
PD-L1 on TILs	0.25 (0.14–0.44)	<0.001	3.42 (1.95–6.01)	<0.001

Hazard ratio, HR; 95% Confidence interval, 95% CI; cancer cells, CCs; tumor-infiltrating lymphocytes, TILs.

**Table 4 medicina-55-00723-t004:** Correlation of PD-L1 expression with clinicopathologic features in patients with colorectal cancer (n = 157).

Parameters	PD-L1 Expression in CCs			PD-L1 Expression on TILs		
	Low n (%)	High n (%)	*p*-Value	Low n (%)	High n (%)	*p*-Value
Sex						
Male	49 (50.5)	48 (49.5)	0.249 †	46 (47.4)	51 (52.6)	0.620 †
Female	36 (60.0)	24 (40.0)		26 (43.3)	34 (56.7)	
Age (median 60)						
(range 19–90)						
19–44	16 (64.0)	9 (36.0)	0.333 ‡	10 (40.0)	15 (60.0)	0.803 ‡
45–54	29 (58.0)	21 (42.0)		24 (48.0)	26 (52.0)	
≥55	40 (48.8)	42 (51.2)		38 (46.3)	44 (53.7)	
Tumor site						
Right side	24 (55.8)	19 (44.2)	0.798 †	21 (48.8)	22 (51.2)	0.648 †
Left side	61 (53.5)	53 (46.5)		51 (44.7)	63 (55.3)	
Tumor size						
<5 cm	37 (61.7)	23 (38.3)	0.138†	25 (41.7)	35 (58.3)	0.410 †
≥5 cm	48 (49.5)	49 (50.5)		47 (48.5)	50 (51.5)	
Histopathologic type						
Adenocarcinoma	57 (57.6)	42 (42.4)	<0.001 ‡	38 (38.4)	61 (61.6)	0.038 ‡
Mucinous	25 (65.8)	13 (34.2)		21 (55.3)	17 (44.7)	
Signet-ring	3 (15.0)	17 (85.0)		13 (65.0)	7 (35.0)	
Histologic grade						
Well	15 (60.0)	10 (40.0)	0.704 ‡	10 (40.0)	15 (60.0)	0.493 ‡
Moderate	59 (54.1)	50 (45.9)		49 (45.0)	60 (55.0)	
Poor	11 (47.8)	12 (52.2)		13 (56.5)	10 (43.5)	
Depth of invasion						
pT1	22 (84.6)	4 (15.4)	<0.001 ‡	7 (26.9)	19 (73.1)	0.026 ‡
pT2	41 (64.1)	23 (35.9)		27 (42,2)	37 (57.8)	
pT3	22 (32.8)	45 (67.2)		38 (56.7)	29 (43.3)	

† Independent simple T-test; ‡ one-way ANOVA; cancer cells, CCs; tumor-infiltrating lymphocytes, TILs.

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
