# Peer review of "Intratumoral Cytotoxic T-Lymphocyte Density and PD-L1 Expression Are Prognostic Biomarkers for Patients with Colorectal Cancer"

_medicina, 2019, doi:10.3390/medicina55110723_

Round 1

Reviewer 1 Report

Authors have performed an interesting work regarding to the potential role of PD-L1 as prognosis marker on CRC. I suggest some minor stylistic changes in order to increase the overall quality of the manuscript: 

Materials and Methods section should be divided on different subsections, one for each procedure e.g. 2.2 Inmunochemistry.

In my opinion, Discussion section can be improved with a minor re-organization of the text. All information regarding to the state-of-art revision should be placed together and before the obtained results. 

Author Response

Response to Reviewer 1 Comments

Point 1: Materials and Methods section should be divided on different subsections, one for each procedure e.g. 2.2 Inmunochemistry.

Response 1: The Materials and Methods section is divided into different subsections, taking your advice into consideration.

Point 2: In my opinion, Discussion section can be improved with a minor re-organization of the text. All information regarding to the state-of-art revision should be placed together and before the obtained results.

Response 2: The discussion section was developed with a comprehensive reorganization of the text. Up-to-date additional information was added in the text and tried to be given before the results obtained.

Reviewer 2 Report

Abstract

The first sentence under results needs to be broken up so it is understandable which outcome is associated with which p value.

You have two sentences about DFS contradicting each other. Is one for cancer cell PD-L1? If so, you need to say this.

Throughout text/Introduction

References often don’t match text. Please correlate references with text.

Methodology

Methods do not describe how PD-L1 was assessed in TILs despite the result being presented in results section and abstract. Please clarify.

Results

The formatting of tables requires some work. Currently some columns are bolded whilst others are not.

Discussion

A lot of text is spent on discussing PD-L1 expression in other cancer types whilst recent papers showing correlation with CRC (including outcomes) are not included. I would suggest a re-write of the discussion to make it more relevant to colorectal cancer specifically.

Overall

My main concerns are the reference discrepancies and the lack of consistency around PD-L1 staining in TILs and cancer cells.

Also do you have microsatellite data on your samples? This would be great to include if you have it.

Author Response

Response to Reviewer 2 Comments

Point 1: Abstract: The first sentence under results needs to be broken up so it is understandable which outcome is associated with which p value. You have two sentences about DFS contradicting each other. Is one for cancer cell PD-L1? If so, you need to say this.

Response 1: Abstract: The first sentence under the results has been rearranged in accordance with your warning and tried to make it more understandable. Corrected conflicting sentences about DFS to eliminate confusion.

Point 2: Throughout text/Introduction: References often don’t match text. Please correlate references with text.

Response 2: Some of the references that did not match the main text were removed from the list. Instead, text-compatible articles were added.

Point 3: Methodology. Methods do not describe how PD-L1 was assessed in TILs despite the result being presented in results section and abstract. Please clarify.

Response 3: Methodology: The method of evaluating PD-L1 expression in TILs was added to the text. PD-L1 expression in TILs is the same as that of expression in cancer cells.

Point 4: Results: The formatting of tables requires some work. Currently some columns are bolded whilst others are not.

Response 4: Results: Tables have been rearranged taking into account your warning.

Point 5: Discussion: A lot of text is spent on discussing PD-L1 expression in other cancer types whilst recent papers showing correlation with CRC (including outcomes) are not included. I would suggest a re-write of the discussion to make it more relevant to colorectal cancer specifically.

Response 5: Discussion: Considering your warnings, the discussion part of the article has been comprehensively modified and re-written. The sections emphasizing other cancers were minimized and discussions about colorectal cancers were highlighted.

Point 6: Overall: My main concerns are the reference discrepancies and the lack of consistency around PD-L1 staining in TILs and cancer cells. Also do you have microsatellite data on your samples? This would be great to include if you have it.

Response 6: Overall: The references conflicts with the text have been eliminated. In the methodology part, it was stated that PD-L1 expression in TILs and cancer cells was similarly evaluated semi-quantitatively. Unfortunately, there is no MSI data for our cases. Therefore, information about these data was not given in our study. I would like to inform you that we will take this precious warning into consideration in the future researches about colorectal cancers.